# Different Mechanisms of Action of Regorafenib and Lenvatinib on Toll-Like Receptor-Signaling Pathways in Human Hepatoma Cell Lines

**DOI:** 10.3390/ijms21093349

**Published:** 2020-05-09

**Authors:** Reina Sasaki, Tatsuo Kanda, Mariko Fujisawa, Naoki Matsumoto, Ryota Masuzaki, Masahiro Ogawa, Shunichi Matsuoka, Kazumichi Kuroda, Mitsuhiko Moriyama

**Affiliations:** Division of Gastroenterology and Hepatology, Department of Medicine, Nihon University School of Medicine, 30-1 Oyaguchi-kamicho, Itabashi-ku, Tokyo 173-8610, Japan; reina_sasaki_0925@yahoo.co.jp (R.S.); mema17020@g.nihon-u.ac.jp (M.F.); matsumoto.naoki@nihon-u.ac.jp (N.M.); masuzaki.ryota@nihon-u.ac.jp (R.M.); ogawa.masahiro@nihon-u.ac.jp (M.O.); matsuoka.shunichi@nihon-u.ac.jp (S.M.); kuroda.kazumichi@nihon-u.ac.jp (K.K.); moriyama.mitsuhiko@nihon-u.ac.jp (M.M.)

**Keywords:** c-JUN, c-FOS, CXCL10, gene expression, HCC, innate immunity, lenvatinib, regorafenib, sorafenib, toll-like receptor

## Abstract

Multiple kinase inhibitors are available for patients with advanced hepatocellular carcinoma (HCC). It is largely unknown whether regorafenib or lenvatinib modulates innate immunity including Toll-like receptor (TLR)-signaling pathways in HCC. We performed real-time RT-PCR to investigate 84 TLR-associated gene expression levels and compared these gene expression levels in each hepatoma cells treated with or without regorafenib or lenvatinib. In response to regorafenib, nine and 10 genes were upregulated in Huh7 and HepG2 cells, respectively, and only C-X-C motif chemokine ligand 10 was upregulated in both cell lines. A total of 14 and 12 genes were downregulated in Huh7 and HepG2 cells, respectively, and two genes (Fos proto-oncogene, AP-1 transcription factor subunit, and ubiquitin conjugating enzyme E2 N) were downregulated in both cell lines. In response to lenvatinib, four and 16 genes were upregulated in Huh7 and HepG2 cells, respectively, and two genes (interleukin 1 alpha and TLR4) were upregulated in both cells. Six and one genes were downregulated in Huh7 and HepG2, respectively, and no genes were downregulated in both cell lines. In summary, regorafenib and lenvatinib affect TLR signaling pathways in human hepatoma cell lines. Modulation of TLR signaling pathway may improve the treatment of HCC patients with refractory disease.

## 1. Introduction

A systematic analysis for the Global Burden of Disease Study demonstrated that there were 953,000 incident cases of liver cancer and 819,000 related deaths in 2017 worldwide [1]. Hepatocellular carcinoma (HCC) is the most common type of liver cancer. It is not easy to detect the early stage of HCC, which is treated by hepatic resection, because even patents with advanced stage HCC show few symptoms [2,3]. Therefore, the development of treatments for HCC is an important issue.

HCC is a heterogeneous tumor with multiple epigenetic and genetic abnormalities [4,5]. Aberrant activation of several signaling cascades such as epidermal growth factor receptor (EGFR), Ras/extracellular signal-regulated kinase, phosphoinositol 3-kinase/mammalian target of rapamycin (mTOR), hepatocyte growth factor/mesenchymal-epithelial transition factor, Wnt, Hedgehog, and apoptotic signaling is often observed [6]. A multikinase inhibitor, sorafenib, has shown survival benefits in patients with advanced HCC [7,8]. Sorafenib has been approved as a first-line systemic chemotherapy for more than 10 years. After the use of sorafenib, the majority of advanced HCC patients are resistant to sorafenib [9]. We and others have reported that overexpression and phosphorylation of Jun proto-oncogene AP-1 transcription factor subunit (c-Jun) contributes to sorafenib resistance [9,10]. The biomarker companion study BIOSTORM showed that hepatocytic pERK and microvascular invasion predicted poor recurrence-free survival [11]. Thus, it is possible that several biomarkers may predict sorafenib benefit in HCC patients.

Regorafenib is one of the approved second-line kinase inhibitors shown to provide survival benefit in patients with HCC who have progressed during sorafenib treatment in Japan and other countries [12]. Lenvatinib was noninferior to sorafenib in terms of overall survival in untreated advanced HCC and is currently an approved kinase inhibitor for the treatment of advanced HCC [13]. Immune checkpoint inhibitors were first shown to be effective in melanoma and lung cancer. Recently immune checkpoint inhibitors have been applied in HCC treatment [14]. There have been several phase III clinical trials, including not only monotherapy with checkpoint inhibitors, programmed death-1 (PD-1) antibody nivolumab, PD-ligand 1 (PD-L1) antibody pembrolizumab, and cytotoxic T-lymphocyte-associated antigen 4 (CTLA-4) antibody tislelizumab, but also on combination therapy with nivolumab or pembrolizumab plus a molecular targeted agent (bevacizumab) or tislelizumab [14,15,16,17,18].

Although monoimmunotherapy is effective in only ~20% of patients with solid cancer [19], Toll-like receptor (TLR) adjuvants may be useful for cancer immunotherapy [20]. TLR signaling pathways in HCC may modulate the treatment outcome of multiple kinase inhibitors with or without cancer immunotherapy. In the present study, we investigated whether regorafenib or lenvatinib modulates innate immunity including TLR-signaling pathways in human hepatoma cell lines. We observed that regorafenib upregulates C-X-C motif chemokine ligand 10 (CXCL10) and downregulates Fos proto-oncogene, AP-1 transcription factor subunit (c-FOS), and ubiquitin conjugating enzyme E2 N (UBE2N) and that lenvatinib upregulates interleukin 1 alpha (IL1A) and TLR4 in both Huh7 and HepG2 cells. Modulation of the TLR signaling pathway may improve the treatment of HCC patients with refractory disease. In the present study, we examined the mechanistic role of regorafenib and lenvatinib on cell proliferation and immunity of human hepatoma cell lines.

## 2. Results

### 2.1. Effects of Regorafenib or Lenvatinib on Cell Viability in Human Hepatoma Cell Lines

In the human body, the blood concentrations of sorafenib, regorafenib, and lenvatinib are ~10 μM, 2 μM, and 0.07–0.1 μM, respectively [9,21,22]. To examine the effects of these drugs on human hepatoma cell lines, Huh7 and HepG2 cells were treated with or without 1, 2, 5, 10, or 20 μM sorafenib, regorafenib, or lenvatinib for 48 h, and cell viability was measured by MTS assay. In Huh7 cells, more than 2 μM sorafenib, regorafenib, and lenvatinib significantly altered cell viability, compared to that in the untreated control (Figure 1). In HepG2 cells, more than 5 μM sorafenib or regorafenib, or 20 μM lenvatinib significantly altered cell viability, compared to that in the untreated control (Figure 2).

We also examined the cell viabilities of Huh7 and HepG2 treated with or without 2 or 20 μM regorafenib or lenvatinib for 48 h, using trypan blue dye exclusion assay. In Huh7 cells treated with 0, 2, or 20 μM regorafenib, cell viabilities (%) were 98.1 ± 1.6, 97.1 ± 0.8 (no statistical difference, compared with untreated control, *n* = 3), and 22.6 ± 7.4 (*p* < 0.05, compared with untreated control, *n* = 3), respectively. In Huh7 cells treated with 0, 2, or 20 μM lenvatinib, cell viabilities (%) were 98.1 ± 1.6, 94.0 ± 3.6 (no statistical difference, compared with untreated control, *n* = 3), and 3.3 ± 0.5 (*p* < 0.05, compared with untreated control, *n* = 3), respectively (Figure 1D–H). In HepG2 cells treated with 0, 2, or 20 μM regorafenib, cell viabilities (%) were 98.6 ± 0.3, 99.0 ± 1.7 (no statistical difference, compared with untreated control, *n* = 3), and 16.7 ± 28.9 (*p* < 0.05, compared with untreated control, *n* = 3), respectively. In HepG2 cells treated with 0, 2, or 20 μM lenvatinib, cell viabilities (%) were 98.6 ± 0.3, 97.9 ± 0.7 (no statistical difference, compared with untreated control, *n* = 3), and 29.6 ± 8.2 (*p* < 0.05, compared with untreated control, *n* = 3), respectively (Figure 2D–H). Because of no effects of 2 μM regorafenib or 2 μM lenvatinib on cell viabilities were observed, we used these concentrations of both drugs for the PCR array experiment.

### 2.2. Effects of Regorafenib on the Toll-Like Receptor (TLR) Signaling Pathway in Human Hepatoma Cell Lines

It is possible that multiple kinase inhibitors, such as regorafenib and lenvatinib, have effects on innate immunity, including the TLR pathway. TLR ligands have been used as adjuvants for traditional vaccines, and they may also play a role in enhancing the efficacy of tumor immunotherapy [19,20,23]. In this study, the effects of multiple kinase inhibitors on innate immunity, including the TLR signaling pathway, were examined in human hepatoma cell lines.

We examined TLR-related gene expression profiles using a real-time PCR-based focused array to investigate the effects of regorafenib on the TLR signaling pathway in Huh7 cells and HepG2 cells. A comparison of TLR-related genes between regorafenib-treated and untreated control Huh7 or HepG2 cells after 24 h of treatment with 2 μM regorafenib is shown in Figure 3. Out of 84 TLR-related genes examined, nine (10.7%) were significantly upregulated in Huh7 cells treated with regorafenib compared to control cells (*n* = 3, *p* < 0.05), while 10 (11.9%) were significantly upregulated in HepG2 cells treated with regorafenib compared to control cells (*n* = 3, *p* < 0.05). The response to regorafenib revealed eight genes unique to Huh7 cells and nine unique to HepG2 cells; only CXCL10 was upregulated in both human hepatoma cell lines. Two genes (CXCL10 and TLR3) were upregulated two-fold or more in Huh7 cells (*n* = 3, *p* < 0.05) and two genes (CXCL10 and BTK) were upregulated two-fold or more in HepG2 cells (*n* = 3, *p* < 0.05). The significantly upregulated genes are summarized in Figure 3A.

On the other hand, out of 84 genes, 14 (16.7%) were significantly downregulated in Huh7 cells treated with regorafenib compared to control cells (*n* = 3, *p* < 0.05), while 12 (14.3%) were significantly downregulated in HepG2 cells treated with regorafenib compared to control cells (*n* = 3, *p* < 0.05). Among them, 12 genes were unique to Huh7, 10 were unique to HepG2 and two genes (FOS and UBE2N) were downregulated in both cell lines. Three genes (FOS, TLR4 and IL1B) were downregulated two-fold or more in Huh7 cells (*n* = 3, *p* < 0.05) and only FOS was downregulated five-fold or more in Huh7 cells (*n* = 3, *p* < 0.05). No genes were downregulated two-fold or more in HepG2 cells (*n* = 3, *p* < 0.05). The significantly downregulated genes are summarized in Figure 3B. Figure 3C,D demonstrate three-dimensional profile of TLR-related gene expression by the treatment of regorafenib in Huh7 and HepG2, respectively.

### 2.3. Effects of Lenvatinib on the Toll-Like Receptor (TLR) Signaling Pathway in Human Hepatoma Cell Lines

Next, we examined TLR-related gene expression profiles using a real-time PCR-based focused array to investigate the effects of lenvatinib on the TLR signaling pathway in Huh7 cells and HepG2 cells. A comparison of TLR-related genes between lenvatinib-treated and untreated control Huh7 or HepG2 cells after 24 h of treatment with 2 μM lenvatinib is shown in Figure 4. Out of 84 TLR-related genes examined, four genes (4.8%) were significantly upregulated in Huh7 cells treated with lenvatinib compared to control cells (*n* = 3, *p* < 0.05), while 16 (19.0%) were significantly upregulated in HepG2 cells treated with lenvatinib compared to control cells (*n* = 3, *p* < 0.05). The response to lenvatinib revealed two genes unique to Huh7 cells, 14 unique to HepG2 cells, and two genes (IL1A and TLR4) upregulated in both human hepatoma cell lines. Only BTK was upregulated two-fold or more in Huh7 cells (*n* = 3, *p* < 0.05). Seven genes (CXCL10, CSF2, IL6, TLR4, LY96, IRAK2 and IL1B) were upregulated two-fold or more in HepG2 cells (*n* = 3, *p* < 0.05), and only CXCL10 was upregulated five-fold or more in HepG2 cells (*n* = 3, *p* < 0.05). The significantly upregulated genes are summarized in Figure 4A.

On the other hand, out of 84 genes, six (7.1%) were significantly downregulated in Huh7 cells treated with lenvatinib compared to control cells (*n* = 3, *p* < 0.05), while only PTGS2 (1.2%) was significantly downregulated in HepG2 cells treated with lenvatinib compared to control cells (*n* = 3, *p* < 0.05). Among them, six genes were unique to Huh7, one was unique to HepG2, and no genes were downregulated in either cell line. No genes were downregulated two-fold or more in Huh7 (*n* = 3, *p* < 0.05) or HepG2 (*n* = 3, *p* < 0.05) cells. The significantly downregulated cells are summarized in Figure 4B. Figure 4C,D demonstrate three-dimensional profile of TLR-related gene expression by the treatment of lenvatinib in Huh7 and HepG2, respectively.

### 2.4. Ingenuity Pathway Analysis (IPA) with Five Genes (CXCL10, FOS, UBE2N, IL1A, and TLR4)

We found five genes (CXCL10, FOS, UBE2N, IL1A, and TLR4) that were significantly changed by regorafenib or lenvatinib in both Huh7 and HepG2 cells. Furthermore, IPA was performed to show that both cell lines and TLR related pathways were prominently influenced by regorafenib and lenvatinib. IPA was applied with a particular focus on the details of the underlying algorithms, and the application to a number of real-world use cases [24]. We performed IPA with these five genes to predict downstream effects and identify new targets or candidate biomarkers (Figure 5). IPA revealed that these five genes that are commonly regulated by both regorafenib and lenvatinib; moreover, they are closely associated with cell proliferation signaling pathways, supporting the results that these two drugs have inhibitory effects on cell proliferation (Figure 1 and Figure 2).

## 3. Discussion

Sorafenib was the first systemic therapy approved for patients with advanced-stage HCC, after the SHARP trial revealed an improvement in median overall survival from 7.9 to 10.7 months [7]. Currently, regorafenib, as a second-line treatment, and lenvatinib have been approved and demonstrated to improve the clinical outcomes of patients with advanced HCC. However, the median overall survival remains ~1 year with these new drugs being used in most patients with advanced HCC [12,13]. Thus, novel therapeutic strategies are still needed to extend the lives of patients with advanced HCC. In the present study, we investigated the molecular mechanisms of lenvatinib and regorafenib in the innate immunity of human hepatoma cell lines to define the effect of these anticancer agents and to support the investigation of novel therapeutic strategies.

We observed significant upregulation of CXCL10 expression in both Huh7 and HepG2 cells treated with regorafenib compared to untreated cells (Figure 3A). CXCL10 is associated with interferon regulatory factor (IRF) signaling and enhances the migration, invasion, and metastasis of HCC cells by activating matrix metallopeptidase-2 (MMP-2) expression [25]. c-FOS and UBE2N expression were significantly downregulated in both cells treated with regorafenib compared to untreated cells (Figure 3B). Interestingly, regorafenib downregulated c-FOS expression, resulting in the suppression of AP-1 activation, which is occasionally seen in sorafenib-resistant HCC [9,10]. Therefore, regorafenib may be effective for treating sorafenib-resistant HCC. UBE2N is related to NF-κB signaling and is required for the development of breast cancer metastasis [26].

There was significant upregulation of IL1A and TLR4 expression in both Huh7 and HepG2 cells treated with lenvatinib compared to untreated cells (Figure 4A). IL1A, which is related to cytokine signaling downstream of TLR, has emerged as an apical driver of inflammation and cancer in the colon [27] and is also associated with distant metastases and poor survival rates in head and neck squamous cell carcinoma and gastric carcinoma [28,29]. TLR4, which has an important role in regulating innate and adaptive immune responses, is expressed in HCC cells and promotes HCC cell proliferation [30,31].

Lipopolysaccharide (LPS) was one of the TLR4 ligands. LPS induced apoptosis of HepG2 was inhibited by decreasing expression of TLR4, associated with hepatitis C virus infection [30]. We observed the downregulation of TLR4 mRNA in HepG2 cells treated with 2 μM regorafenib and the upregulation of TLR4 mRNA in HepG2 cells treated with 2 μM lenvatinib (Figure 4C,D). Although we also examined whether 100 ng/mL LPS could enhance the effects of 2 μM regorafenib or 2 μM lenvatinib on cell death, we did not observe any differences of cell death between two drugs (data not shown).

Our findings for IL1A and TLR4 mRNA upregulation induced by lenvatinib is important (Figure 4A). It could be valid for the immune cells although our experimental study was performed in two human hepatoma cell lines. Since interleukin-1 receptor and TLR4 signaling triggers MYD88 innate immune signal transduction adaptor (MyD88) activation cascade in monocytes, the effect of levantinib could be not only on hepatoma cells, but also on monocytes. Our results could support the basis of concept that lenvatinib, a first line drug for HCC, synergizes with immunotherapy to inhibit the immunosuppressive tumor environment, via the inhibition of the monocytes to tumor-associated macrophages transition or the inhibition of vascular endothelial growth factors as the main mechanism [32,33]. Further study is needed.

A human phase II clinical trial using an anti-CXCL10 monoclonal antibody (MDX-1100) for rheumatoid arthritis patients who had an inadequate response to methotrexate treatment showed that blocking CXCL10 significantly increased the response rate compared to that in the placebo group, suggesting a possible therapeutic use in humans [34]. MABp1 is a human antibody targeting IL1A and is undergoing clinical trials for moderate to severe hidradenitis suppurativa patients not eligible for adalimumab [35]. TLR4 inhibitors (eritoran and resatorvid) and TLR4 agonists (GSK1795091, OM174) are also under clinical trials [36,37,38,39]. New multiple kinase inhibitors, such as ramucirumab and cabozantinib, and monoimmunotherapy are available or will be available soon. Drugs against these molecules may enhance the effects of multikinase inhibitors on HCC.

## 4. Materials and Methods

### 4.1. Cell Culture and Regents

Human hepatoma cell lines HepG2 and Huh7 were used as previously described [9,40]. Cells were maintained in Roswell Park Memorial Institute medium (RPMI1640) (Sigma-Aldrich, St. Louis, MO, USA) supplemented with 10% fetal calf serum (FCS) and 200 U/mL penicillin and 200 μg/mL streptomycin at 37 °C in a 5% CO_2_ atmosphere. Sorafenib, regorafenib, and lenvatinib were purchased from AdooQ Bioscience (Irvine, CA, USA), Cayman Chemical (Ann Arbor, MI, USA), and ChemScene LLC (Monmouth Junction, NJ, USA), respectively.

### 4.2. MTS Assay and Trypan Blue Dye Exclusion Assay

Cells were treated with sorafenib, regorafenib, or lenvatinib at 0, 1, 2, 5, 10, or 20 μM for 48 h. To determine cell viability, the CellTiter 96 AQueous One Solution Cell Proliferation Assay (Promega, Madison, WI, USA) was used [9]. Living cells converted 5-(3-carboxymethoxyphenyl)-2-(4,5-dimenthylthiazoly)-3-(4-sulfophenyl) tetrazolium, inner salt (the MTS tetrazolium compound) to formazan. To perform the MTS assay, MTS solution was added to each well. Four hours later, the absorbance of each cells at 490 nm was measured with the iMark Microplate Absorbance Reader (Bio-Rad, Tokyo, Japan).

Cells were treated with regorafenib or lenvatinib at 0, 2, or 20 μM for 48 h. Cell pictures were also taken by phase contrast microscopy (BIOREVIO BZ-9000, Keyence, Osaka, Japan). Cell suspension were also stained with 0.4% trypan blue dye (Invitrogen, Carlsbad, CA, USA), and both live and dead cells were counted under phase contrast microscopy [41].

### 4.3. RNA Extraction, cDNA Synthesis and Human TLR Signaling Targets PCR Array

Twenty-four hours before treatment, approximately 0.5 × 10^6^ cells were plated into each well of six-well plates. Cells were treated with or without 2 μM regorafenib or 2 μM lenvatinib for 24 h and cellular RNA was extracted using the RNeasy Mini Kit (Qiagen, Hilden, Germany). RNA was quantified by using a NanoDrop ND-1000 spectrophotometer (Thermo Fisher, Tokyo, Japan), and 0.5 μg RNA was subjected to each reaction. cDNA was synthesized with an RT^2^ First Strand cDNA Kit (Qiagen) for 15 min at 42 °C, for 5 min at 95 °C, and for min at 72 °C. A human TLR signaling pathway PCR array was purchased from Qiagen. The 84 genes associated with the TLR signaling pathway assessed by real-time RT-PCR are listed in Appendix A [42]. A real-time PCR array based on the SYBR Green method was performed on a 7500 Fast real-time PCR system (Applied Biosystems, Foster, CA, USA). The cycling program was as follows: 95 °C for 10 min for one cycle, then 40 cycles of 95 °C for 15 s and 60 °C for 1 min. The relative gene expression levels were normalized to five housekeeping gene levels by using the 2−ΔΔ*C*T formula (ΔΔ*C*T = Δ*C*T sample—Δ*C*T untreated control). Data were analyzed by the RT2 Profiler PCR Array Data Analysis software (http://pcrdataanalysis.sabiosciences.com/pcr/arrayanalysis.php). The analysis of potential target genes was carried out using IPA (Qiagen). Gene expression changes were considered statistically significant at *p* < 0.05.

### 4.4. Statistical Analysis

The results are presented as the means ± standard deviations of triplicate determinations from one experiment, representative of three independent experiments. Data were analyzed by Student’s *t* test with a two-tailed distribution. A *p*-value of < 0.05 was considered statistically significant.

## 5. Conclusions

Regorafenib and lenvatinib have effects on TLR signaling pathways in human hepatoma cell lines (Appendix A). Regorafenib and lenvatinib also have different actions in innate immunity. However, there exist some limitations in this study. First, as we used only two hepatoma cell lines; we should confirm the results in an in vivo study. Second, although we observed the change of gene expression levels, we did not see the protein expression levels or downstream signaling mechanism. Third, although hepatocyte is one of the immune cells [43], we did not examine the interaction between hepatocytes and other immune cells [32,33,44,45,46].

Our study is limited, but it is the beginning of our project. These limitations could be addressed in future research. Regorafenib downregulates c-FOS expression, resulting in the suppression of AP-1 activation, which is occasionally seen in sorafenib-resistant HCC [9,10]. Innate immune response pathways play important roles in exerting the effects of molecularly targeted drugs against HCC. Modulation of the TLR signaling pathway may improve the treatment of HCC patients with refractory disease.

## Figures and Tables

**Figure 1 ijms-21-03349-f001:**
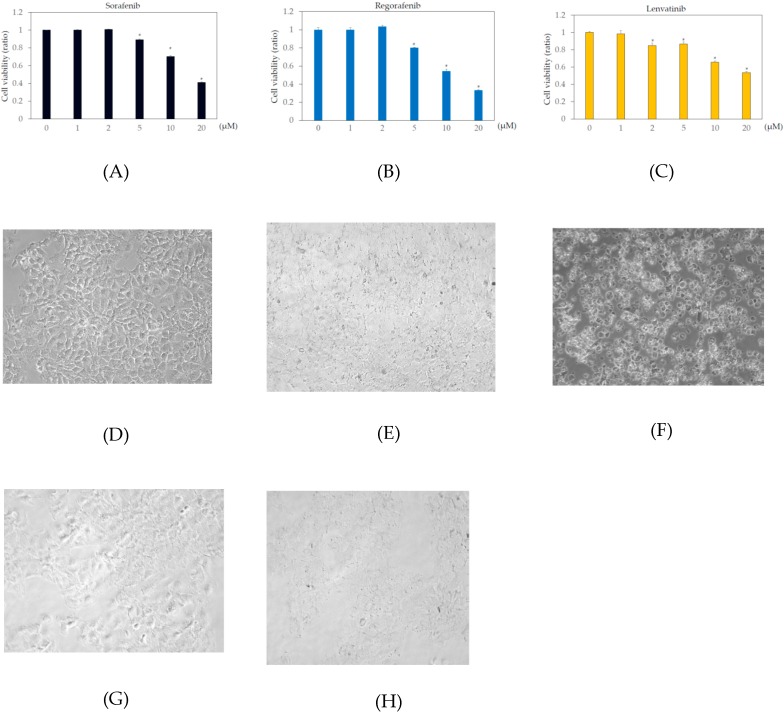
Effects of multiple kinase inhibitors on the Huh7 cell viability cells (*n* = 3), (**A**) sorafenib; (**B**) regorafenib; (**C**) lenvatinib. Huh7 cells were treated with sorafenib, regorafenib or lenvatinib at 0, 1, 2, 5, 10, or 20 μM for 48 h. Cell viability was measured by MTS assay. * *p* < 0.05, compared to Huh7 treated without multiple kinase inhibitors. Pictures taken by phase contrast microscopy (×20): (**D**) Huh7 control; (**E**) Huh7 cells treated with 2 μM regorafenib; (**F**) 20 μM regorafenib; (**G**) 2 μM lenvatinib; (**H**) 20 μM lenvatinib.

**Figure 2 ijms-21-03349-f002:**
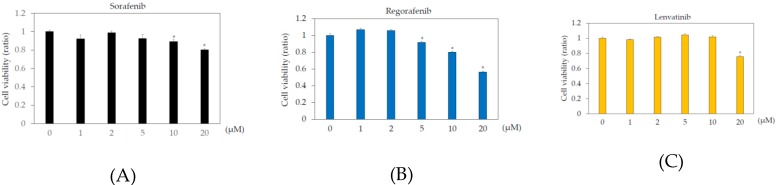
Effects of multiple kinase inhibitors on the HepG2 cell viability cells (*n* = 3), (**A**) sorafenib; (**B**) regorafenib; (**C**) lenvatinib. HepG2 cells were treated with sorafenib, regorafenib, or lenvatinib at 0, 1, 2, 5, 10, or 20 μM for 48 h. (**A**) sorafenib; (**B**) regorafenib; (**C**) lenvatinib. Cell viability was measured by MTS assay. * *p* < 0.05, compared to HepG2 treated without multiple kinase inhibitors. Pictures taken by phase contrast microscopy (×20): (**D**) HepG2 control; (**E**) HepG2 cells treated with 2 μM regorafenib; (**F**) 20 μM regorafenib; (**G**) 2 μM lenvatinib; (**H**) 20 μM lenvatinib.

**Figure 3 ijms-21-03349-f003:**
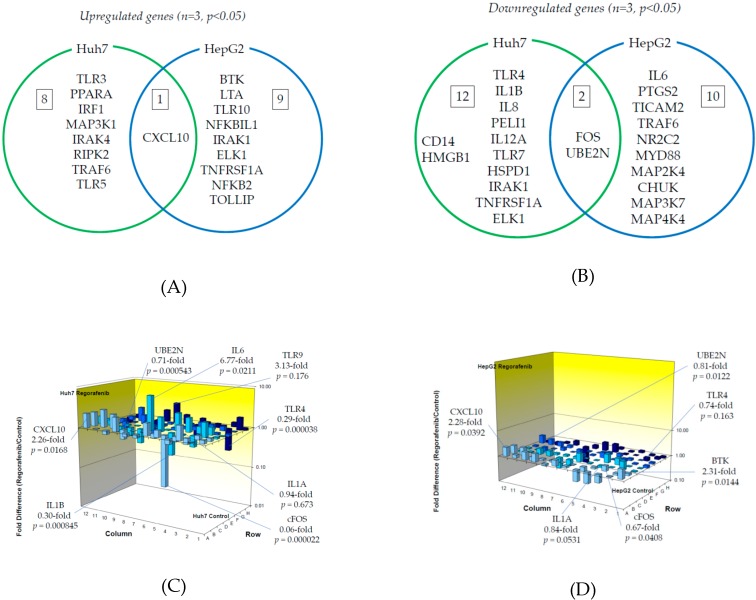
Effects of regorafenib treatment on Toll-like receptor (TLR)-related gene expression in human hepatoma Huh7 and HepG2 cells: (**A**) Upregulated genes (*p* < 0.05); (**B**) Downregulated genes (*p* < 0.05); (**C**) Changes of TLR-related gene expression in Huh7 cells treated with or without regorafenib; (**D**) Changes of TLR-related gene expression in HepG2 cells treated with or without regorafenib. Cells were treated with or without 2 μM regorafenib for 24 h and cellular RNA was extracted. Eighty-four TLR-related genes were evaluated by real-time PCR-based array (*n* = 3). *p*-values, compared to those of untreated cells.

**Figure 4 ijms-21-03349-f004:**
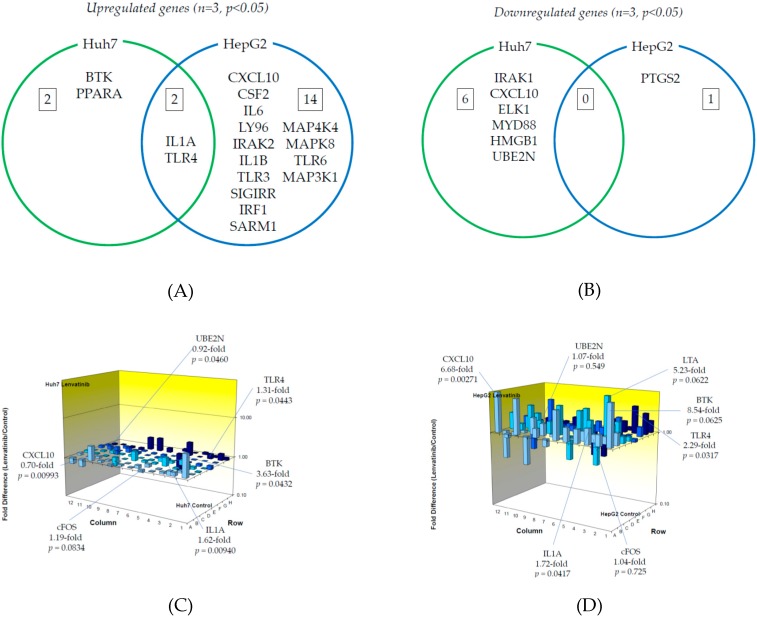
Effects of lenvatinib treatment on Toll-like receptor (TLR)-related gene expression in human hepatoma Huh7 and HepG2 cells: (**A**) Upregulated genes (*p* < 0.05); (**B**) Downregulated genes (*p* < 0.05). (**C**) Changes of TLR-related gene expression in Huh7 cells treated with or without lenvatinib; (**D**) Changes of TLR-related gene expression in HepG2 cells treated with or without lenvatinib. Cells were treated with or without 2 μM lenvatinib for 24 h and cellular RNA was extracted. Eighty-four TLR-related genes were evaluated by real-time PCR-based array (*n* = 3). *p*-values, compared to those of untreated cells.

**Figure 5 ijms-21-03349-f005:**
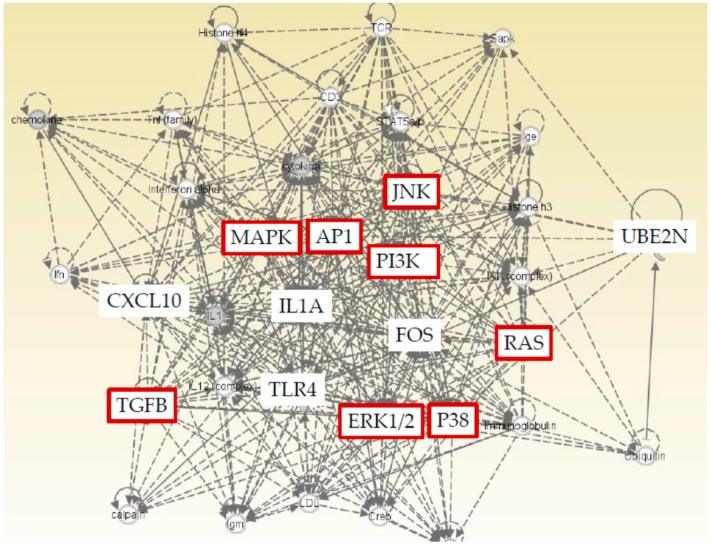
Ingenuity Pathway Analysis (IPA) of five genes (CXCL10, FOS, UBE2N, IL1A, and TLR4) that are commonly regulated by both regorafenib and lenvatinib showed a close association with cell proliferation signaling pathways (red color).

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
