# Peer review of "Different Mechanisms of Action of Regorafenib and Lenvatinib on Toll-Like Receptor-Signaling Pathways in Human Hepatoma Cell Lines"

_ijms, 2020, doi:10.3390/ijms21093349_

Round 1

Reviewer 1 Report

Different mechanisms of action of regorafenib and lenvatinib on Toll-like receptor-signaling pathways in human  hepatoma cell lines has been written by Sasaki Reina et al. They have  described  the effects of 2 kinase inhibitors regorafenib & lenvatinib in vitro.

The authors  improve their paper by following our comments. They understand their study is limited but they explain their study is the beginning of their project. 

Minor concerns: 

-figure 1: scale bar is missing ; homogenize the size of picture H  

-figure 2: homogenize the size of picture D

Author Response

Response to Reviewer 1

Thank you very much for your invaluable comments.

Response to your minor comment 1: “figure 1: scale bar is missing ; homogenize the size of picture H”

Thank you for your valuable comments. Accordingly, we revised Figure 1A, 1B, 1C and 1H.

Response to your minor comment 1: “figure 2: homogenize the size of picture D”

Thank you for your valuable comments. Accordingly, we revised Figure 2.

Reviewer 2 Report

In my original review, additional experiments were requested; however, the authors did not attempt to address these concerns and instead defaulted all comments to the Discussion section as limitations of the study. Indeed, if the authors are unable to fulfill these requests due to money or time restraints, this should have been stated.

Author Response

Response to Reviewer 2

Thank you very much for your comments.

Response to your comments: “In my original review, additional experiments were requested; however, the authors did not attempt to address these concerns and instead defaulted all comments to the Discussion section as limitations of the study. Indeed, if the authors are unable to fulfill these requests due to money or time restraints, this should have been stated.”

Thank you for your valuable comments. As you mentioned, the grant of JSPS KAKENHI Grant Number JP17K09404 (to T.K.) was exhausted at 31 March, 2020. Our study is limited but our study is the beginning of our project. We revised our manuscript as follows.

In Conclusion section, page 9, line 284,

…. and other immune cells [32,33,44-46]. Our study is limited but our study is the beginning of our project. These limitations could be addressed in future research….

Reviewer 3 Report

Results of this work are important in the field of the basis of the concept that levantinib ,a first line drug for HCC, synergizes with immunotherapy to inhibit the immunosupressive tumor microenvironment, currently thought via inhibition of the monocyte to TAM transition (ref 44 of the paper , Kudo M. Liver Cancer 2018;7:20-27),or VEGF inhibition, as the main mechanisms.

Findings for IL1A and TLR4 mRNA upregulation are important ,and although found in this work experimentally in 2 Hepatoma cell lines ,could be valid for the immunocytes also.(speculation)

Since IL1R and TLR4 signaling in monocytes triggers MyD88 activation cascade, the effect of levantinib could be not only on Hepatoma cells ,but also on monocytes.

I would like to see in this paper this speculation in the discussion with a phrase, in the final publication.

Author Response

Response to Reviewer 3

Thank you very much for your invaluable comments.

Response to your comments: “Results of this work are important in the field of the basis of the concept that levantinib ,a first line drug for HCC, synergizes with immunotherapy to inhibit the immunosupressive tumor microenvironment, currently thought via inhibition of the monocyte to TAM transition (ref 44 of the paper , Kudo M. Liver Cancer 2018;7:20-27),or VEGF inhibition, as the main mechanisms.

Findings for IL1A and TLR4 mRNA upregulation are important ,and although found in this work experimentally in 2 Hepatoma cell lines ,could be valid for the immunocytes also.(speculation)

Since IL1R and TLR4 signaling in monocytes triggers MyD88 activation cascade, the effect of levantinib could be not only on Hepatoma cells ,but also on monocytes.”

Thank you for your valuable comments. Accordingly, we added a new reference [32] and revised our manuscript as follows.

In Discussion section, page 7, lines 217-225,

Our findings for IL1A and TLR4 mRNA upregulation induced by lenvatinib is important (Figure 4A). It could be valid for the immune cells although our experimental study was performed in 2 human hepatoma cell lines. Since interleukin-1 receptor and TLR4 signaling triggers MYD88 innate immune signal transduction adaptor (MyD88) activation cascade in monocytes, the effect of levantinib could be not only on hepatoma cells, but also on monocytes. Our results could support the basis of concept that lenvatinib, a first line drug for HCC, synergizes with immunotherapy to inhibit the immunosuppressive tumor environment, via the inhibition of the monocytes to tumor-associated macrophages transition or the inhibition of vascular endothelial growth factors as the main mechanism [32,33]. Further study is needed.

Round 2

Reviewer 2 Report

no comments

This manuscript is a resubmission of an earlier submission. The following is a list of the peer review reports and author responses from that submission.

Round 1

Reviewer 1 Report

Different mechanisms of action of regorafenib and lenvatinib on Toll-like receptor-signaling pathways in human  hepatoma cell lines has been written by Sasaki Reina et al. They have  described  the effects of 2 kinase inhibitors regorafenib & lenvatinib in vitro.

First of all, they studied their toxicity on 2 common cells lines, Huh-7 and HepG2. The figure 1 & 2 have shown the ratio of cell viability either control drug (Sorafenib) or the 2 drug of interest (Regorafenib and Lenvatinib). This figure should improved with  some cell pictures. The histogram should show the statistical difference (this point is mentionned only in the text). 

In the figure 3 and 4, they have done   transcriptome analysis of cells incubated with 2 μM in of Inhibitor of interest  during 24 hours. Some explications should add for the  concentration and the time of the kinase inhibitors. 

For understanding, a heat map is appreciated to see the level of  all genes expression. 

In the figure 5, the authors have built an ingenuity Pathways analysis with Qiagen helphing. They concluded that the 2 inhibitors have effects on cell proliferation. This data is obtained in silico. the conclusion should consolidate by experiments on cell. 

In the discussion, the authors come back on CXCL10, IL1A, TLRR4, c-FOS and UBE2N expression in 2 cells lines. They discuss why these genes are interesting by using the litterature. Some experiments in vitro are missing to  extend the modulation  of these genes by these 2 drugs. 

Reviewer 2 Report

In this manuscript, the authors tried to evaluate the mechanistic role of lenvatinib and regorafenib on HCC cell proliferation and immunity. 

Major comments:

  • The study was only performed in 2 HCC cell lines, others should have been evaluated to show if these findings can be replicated.
  • Work in vivo would have been highly beneficial.
  • While the authors propose different pathways, they do not perform experiments looking at the downstream signaling mechanisms. This is needed to prove that these pathways are actually being manipulated.
  • The authors state that these drugs manipulate innate immunity, but experiments with innate immune cells should have been performed; such as, co-culture studies.
  • Cell proliferation studies should be validated with other experimental methods.
  • The study relies on gene data, but protein expression should have been looked at.
  • Findings from the IPA should be validated.

Minor comments:

  • The figures should be appropriately labeled. As well, they seem to be missing significance markers?